# On the Use of the AIRA-UAS Corpus to Evaluate Audio Processing Algorithms in Unmanned Aerial Systems [note 1]

**DOI:** 10.3390/s19183902

**Published:** 2019-09-10

**Authors:** Caleb Rascon, Oscar Ruiz-Espitia, Jose Martinez-Carranza

**Affiliations:** 1Instituto de Investigaciones en Matematicas Aplicadas y en Sistemas, Universidad Nacional Autonoma de Mexico (UNAM), Mexico City 04510, Mexico; 2Computer Science Department, Instituto Nacional de Astrofísica, Óptica y Electrónica (INAOE), Puebla 72840, Mexico; 3Computer Science Department, University of Bristol, Bristol BS8 1UB, UK

**Keywords:** corpus evaluation, AIRA-UAS, AQBNE, IMCRA, unmanned aerial systems

## Abstract

Audio analysis over an Unmanned Aerial Systems (UAS) is of interest it is an essential step for on-board sound source localization and separation. This could be useful for search & rescue operations, as well as for detection of unauthorized drone operations. In this paper, an analysis of the previously introduced Acoustic Interactions for Robot Audition (AIRA)-UAS corpus is presented, which is a set of recordings produced by the ego-noise of a drone performing different aerial maneuvers and by other drones flying nearby. It was found that the recordings have a very low Signal-to-Noise Ratio (SNR), that the noise is dynamic depending of the drone’s movements, and that their noise signatures are highly correlated. Three popular filtering techniques were evaluated in this work in terms of noise reduction and signature extraction, which are: Berouti’s Non-Linear Noise Subtraction, Adaptive Quantile Based Noise Estimation, and Improved Minima Controlled Recursive Averaging. Although there was moderate success in noise reduction, no filter was able to keep intact the signature of the drone flying in parallel. These results are evidence of the challenge in audio processing over drones, implying that this is a field prime for further research.

## 1. Introduction

Current advances in technology have greatly improved different sectors of the industry and society. One such sector has been the area of aerial vehicles piloted by radio control. Very recently, Unmanned Aerial Systems (UAS) were used exclusively for military purposes, first in the shape of airplanes and helicopters, and later in the shape of multi-rotors. Nowadays, UAS have been implemented in several sectors of society, such as agriculture, surveillance, transportation, environment monitoring, topography, and recreation [1,2].

In many of these developments, a plethora of sensor modalities have been used for different application cases, such as vision [3,4], infrared [5], and even audio [6,7]. In the case of audio, an important application case is that of detecting unauthorized drone operations. For instance, the scenario where UAS should be removed because they are flown in restricted zones, such as exclusive aerial spaces for commercial airplanes and privately-owned areas. Tracking them can be carried out by vision, but it could be greatly improved by the use of sound source localization algorithms, such as the one presented in [7], since audio can be captured omni-directionally.

In addition, one of the major characteristics of an UAS is the sound its rotors produce while flying, which could be used for its detection and tracking in such restricted zones [8,9]. Another scenario of interest is that of carrying out audio signal processing over a drone could be useful in rescue situations [10,11]. However, the high amount of energy of the ego-noise of the drone while flying could be considered a challenge [6] but this has not been studied extensively. This is the main objective of the work presented here: to evaluate if current popular techniques are able to properly filter out the ego-noise of the drone’s motors. To do this it is important to first analyze the characteristics of such a noise and choose filtering techniques appropriately. Having chosen these, a proper evaluation needs to be carried out to measure their performance.

To this effect, it is indispensable to use adequate evaluation resources and methodologies. We previously introduced the Acoustic Interactions for Robot Audition (AIRA)-UAS corpus in [12], and we believe that it is perfectly compatible with these objectives. This corpus consists of audio recordings that were captured over a flying drone. Different recordings settings were used, such as the drone being static at different heights, vertical, and horizontal movements, as well as the presence of other drones. This corpus is named under the umbrella of another corpus made for evaluating audio signal processing algorithms called AIRA [13]. AIRA is aimed for evaluating sound source localization and separation algorithms but in the context of service robotics. The AIRA-UAS corpus is an extension of AIRA in the realm of unmanned aerial systems (UAS), and therefore named as such. The full set of recordings can be obtained from [14].

As it will be seen, there are recordings in AIRA-UAS where another drone is flying in parallel to the capture drone, and they present a very low Signal-to-Noise Ratio (SNR), which is to be expected. Thus, extracting the audio signature of another flying drone is a challenge. Additionally, it will also be seen that the noise spectral signature changes depending if the drone is static or moving. However, in AIRA-UAS the spectral signature of the capture drone is also provided, which may prove useful for its cancellation.

As for which techniques to evaluate, it is important to mention that there is a vast a mount of filtering and noise cancellation techniques available [15]. We decided upon Berouti’s Non-Linear Noise Subtraction [16] (NLNS), Adaptive Quantile Based Noise Estimation [17] (AQBNE), and Improved Minima Controlled Recursive Averaging [18] (IMCRA). We chose these because of their popularity, ease of implementation and, more importantly, the amount of required computing resources is such that they can run out on the drone’s on-board computer. It is important to mention that the adaptive nature of AQBNE and IMCRA is appropriate for this application, since the ego-noise of the drone is dynamic. And, while the NLNS is not adaptive, it can use the drone’s spectral signature as the noise estimation to cancel out.

This paper is organized as follows: Section 2 describes the relevant parts of the AIRA-UAS corpus for completeness sake; Section 3 provides an analysis of the audio recordings in AIRA-UAS from the time-frequency domain and describe the challenges put forward by carrying out audio processing over a flying drone; Section 4 details the filtering techniques to be evaluated; Section 5 describes the evaluation methodology based on the AIRA-UAS recordings, as well as presents the results obtained from such evaluation; and Section 6 offers the work’s conclusions and future work, as well as some ideas for the UAS community.

## 2. The AIRA-UAS Corpus

The AIRA-UAS corpus was previously introduced in [12], and its relevant parts are described here for completeness sake. It was captured with the intent of being used for evaluating audio processing techniques carried over an UAS. Techniques, such as sound source localization and separation, are of interest, but all of them require a pre-processing stage that involves the removal of the drone’s motors ego-noise. To this effect, the AIRA-UAS was designed to provide a varied selection of flying patterns and accompanying flying drones. As mentioned before, it is freely available to download at [14].

### 2.1. Hardware

#### 2.1.1. Unmanned Aircraft Vehicles

In this work, the microphones were mounted on an UAS, and another two drones were used to fly around that UAS and generate additional audio data.

##### DJI Matrice 100

This is the drone where the microphones were mounted. The camera and gimbal were removed to provide space and not impact its weight limitations. It is shown in Figure 1a and has the following specifications:
**Diagonal wheelbase:** 650 mm**Propeller model:** DJI 1345 × 2 (13 in diameter and 4.5 in pitch, 2 blades)**Weight (with TB47D battery model):** 2355 g**Max. Takeoff Weight:** 3400 g

##### 3DR Solo

This drone is shown in Figure 1b, and was one of the two drones that flew around the DJI Matrice 100. It has the following specifications:
**Diagonal wheelbase:** 460 mm**Propeller model:** 1045 × 2 (10 in diameter and 4.5 in pitch, 2 blades)

##### Parrot Bebop 2

This drone is shown in Figure 1c, and was the second of the two that flew around the DJI Matrice 100. It has the following specifications:
**Dimensions:** 38 × 33 × 9 cm**Propeller model:** 5040 × 3 (5 in diameter and 4 in pitch, 3 blades)

#### 2.1.2. Instrumentation

The audio capture hardware is composed by a RaspberryPi 2 running Raspbian OS as the main processing unit, and the 8SoundsUSB audio interface [19] (shown in Figure 2a) with its eight microphones (show in Figure 2b). The reasoning behind using this hardware is because it is lightweight and requires a low amount of energy. At the same time, however, it is able to capture up to 8 channels simultaneously. This amount of microphones is very popular when carrying out multiple sound source localization [20]. The complete system is shown in Figure 2c.

### 2.2. Recording Protocols

Three recording protocols were used, each with its own set of objectives. In this section a detailed explanation of each protocol is provided.

#### 2.2.1. 1st Protocol

The intent for this set of recordings is to capture the background noise and the ego-noise of the DJI Matrice 100 doing several aerial maneuvers. Additionally, several recordings only differ between them in terms of the height that the UAS was flying at, with the intent of exploring if the ego-noise of the UAS is dependent of its height.

There were 15 recordings captured and their relevant details are presented in Table 1.

A brief summary of each recording is provided here:
*Recording 1.* A preliminary test to ascertain that all the systems were running correctly. The drone was not moving and motors were not activated.*Recording 2 and 3.* Background noise recordings, so that spectral filtering could potentially be used.*Recording 4.* Motors activated, but not taken off.*Recording 5, 8, and 11.* Vertical displacements at different heights. These tests were only 15 s, since the displacements were carried out very quickly by the drone.*Recording 6, 9, 12, 14, and 15.* Drone hovering at different heights, but not moving.*Recording 7, 10, and 13.* Horizontal displacements at different heights.

#### 2.2.2. 2nd Protocol

The intent for this set of recordings is to capture the DJI Matrice 100 and the 3DR Solo flying at the same time, as well as obtain the spectral signature of the 3DR Solo from afar. 3 recordings were captured:*Recording 16.* The motors of the DJI Matrice 100 were not activated and the 3DR Solo was flying at a horizontal distance of 2 m and a height of 1 m.*Recording 17.* The motors of the DJI Matrice 100 were not activated and the 3DR Solo was flying around it forming a complete circle with a radius of 2 m at a height of 1 m.*Recording 18.* Both the DJI Matrice 100 and the 3DR Solo were flying in front of each other at a height of 5 m with a distance of 3 m between them.

#### 2.2.3. 3rd Protocol

The intent for this set of recordings is very similar to the one of the 2nd protocol. The only difference is that the 3DR Solo was replaced by the Parrot Bebop 2. The same three recording settings of the 2nd protocol were carried out (Recordings 19, 20, and 21).

## 3. Analysis of AIRA-UAS Recordings

As stated before, it is important to first analyze the characteristics of the recordings in AIRA-UAS to adequately choose the filtering techniques that are going to be evaluated. To this effect, four different parts of the corpus are analyzed:The environment in which the recordings were made.The noise signatures of the drones captured when flying but not moving.The difference in noise signatures when the drones are moving.The recordings when two drones are flying in parallel.

These analyses are carried out mainly by visual inspection of the spectrogram in the time-frequency domain of the first channel of the recordings. For completeness, it is important to note that these spectrograms are calculated with a Hann window size of 1024 samples, with a 50% overlap, and normalized by the window size. In some cases, for ease of visualization, the mean-energy-per-frequency of the spectrogram is also presented.

Additionally, other elements, such as the signal energy and SNR, are also used, as well as subjective hearing in some cases. Finally, it is important to mention that the data here shown are of the sections that are relevant to the analysis; the data shown in the website of the AIRA-UAS is only representative of the movements for each recording and may not exactly coincide with the data shown here.

### 3.1. Recording Environment

The recording used for this part was the Recording 2 of AIRA-UAS, when the Matrice is not active and only the environment was captured. A mean energy value of -61.657 dBFS was measured, which implies that there was a very low amount of environmental noise.

The spectrogram of this recording is shown in Figure 3a and a zoomed version is presented in Figure 3b. It is important to mention that the same color-map is used in all spectrograms.

As it can be seen, although there is not much energy in most of the frequency throughout the recordings, there is a small amount in the range below 5 kHz.

This implies that the system was not well grounded, as it appears that there is some electrical noise, which was confirmed by subjective hearing of the recording. However, as it will be seen in the following sections, this amount of noise is still quite low compared to the recording of the active drone motors.

### 3.2. Drone Signatures

The recordings used for this part were Recordings 4 (with active motors but not flying) and 6 (flying in a static position) for the Matrice, and Recordings 16 and 19 for the 3DR and Parrot, respectively.

In Figure 4a,b the spectrograms of Recordings 4 and 6 are shown.

As it can be seen, there is a difference of energy presence in the range between 10 and 15 kHz between these two recordings. Additionally, there is an important increase in energy in the range between 2.5 kHz and 7.5 kHz, which can be visualized in the mean-energy-per-frequency of Recordings 4 and 6 in Figure 5a,b.

The increase in energy from Recording 4 to 6 is expected, since the motors create more noise when flying than just laying on the ground.

It could be argued that Figure 5b shows the Matrice noise signature in the frequency domain. The noise signatures of the 3DR and the Parrot are shown in Figure 6a,b.

As it can be seen, the Matrice, the 3DR and the Parrot bare different noise signatures. This implies that the idea of recognizing a drone by its noise signature may have some merit. However, this will only be possible if the ego-noise of the Matrice is filtered out. Comparing the three signatures, there is a significant difference in energy (almost two orders of magnitude) between the signatures, which implies a severely low SNR. Additionally, the correlation between the three signatures is shown in Table 2.

As it can be seen, the correlation factors between the three drone noise signature are quite close; thus, their extraction from a recording of two drones flying in parallel is very challenging. However, this will be discussed further in Section 3.4.

### 3.3. Drone Displacement

A recording used for this part was Recording 6 (as already shown in Figure 4b) to show when the Matrice is flying in a static point. Additionally, the Recordings 5 (taking off), 8 (vertical displacement), and 7 (horizontal displacement) are also used.

Because the objective of this part of the analysis is to show how dynamic the ego-noise of a drone is, a zoomed version of the spectrograms of Recordings 5 and 6 are shown in Figure 7a,b.

As it can be seen, taking off (which implies a change in velocity) versus moving in a constant velocity exemplifies how the ego-noise of the drone changes when its velocity changes.

In Figure 8a,b the full spectrogram of Recordings 8 and 7 are shown.

Compared to the spectrogram of Recording 6 (flying in a static position) shown in Figure 4b, small variations of energy can be seen throughout the displacement recordings. To better present these changes, the frequency-wise average of the energy variance per frequency is shown in Table 3.

Although there is an increase of energy variance when the drone is moving compared to when it is static, further statistical analysis is required to prove this was because of the ego-noise and not other factors, such as wind, sensor noise, and the accuracy of automatic control algorithm. However, an interesting preliminary observation is that a horizontal displacement presents a much greater energy variance than a vertical displacement. This could be attributed to the fact that a vertical displacement involves the same amount of velocity change in all the motors and a horizontal displacement involves different velocity changes in different motors.

This part of the analysis shows a tendency between the changes in motor activity and the noise captured, which is to be expected. However, this also raises the issue of the challenge of filtering the ego-noise, because it is non-stationary. This points to the necessity of requiring adaptive filtering techniques to remove the ego-noise of the drone.

### 3.4. Drones Flying in Parallel

In this part of the analysis, the mean-energy-per-frequency of the spectrogram Recording 6 (as already shown in Figure 5b) was considered as the noise signature of the Matrice when flying alone. Additionally, Recordings 18 (the Matrice and the 3DR flying in parallel) and 21 (the Matrice and the Parrot flying in parallel) were also used.

In Figure 9a,b the mean-energy-per-frequency of the spectrogram of Recordings 18 and 21 are shown.

If we visually compare these figures to Figure 5b which presents the mean-energy-per-frequency of the spectrogram of Recording 6 of the Matrice, there is no clear difference between the three. In fact, it would be a near impossible task to pinpoint visually where the influence of the 3DR noise signature (Figure 6a) and of the Parrot noise signature (Figure 6b) are in Figure 9a,b, respectively. This is because, as mentioned earlier, the energy captured from the noise of the Matrice compared to the noise captured from the 3DR and the Parrot differs in several orders of magnitude. This presents a big challenge in terms of the severely low SNR that should be expected in this application study.

However, when carrying out a subjective hearing of Recording 18, it is somewhat possible to pick up the presence of the 3DR by comparing it to the recording of the Matrice alone. Thus, this points to a possibility that noise cancellation may be viable.

## 4. Filtering Techniques

As mentioned in Section 3, AIRA-UAS provides the noise signature of the three drones in question; thus, noise subtraction techniques should be considered. Additionally, because the noise is dynamic (depending on the motor activity of the drone), adaptive filtering techniques should also be evaluated. To this effect, the following three popular filtering techniques are to be evaluated and are briefly described in this section:
Berouti’s Non-Linear Noise Subtraction [16]Adaptive Quantile Based Noise Estimation [17]Improved Minima Controlled Recursive Averaging [18]

In the following descriptions, the signal model presented in Equation (Equation 1) is assumed.
(1)X=S+N
where *X* is the recorded signal; *S* is the signal of interest; and *N* is the noise to be removed or filtered out. All signals are assumed to be in the frequency domain.

### 4.1. Non-Linear Noise Subtraction (NLNS)

Typically, once an estimation of the noise signal in the frequency domain ($\hat{N}$) is obtained, filtering it in the frequency domain should be just a matter of subtracting from the recorded signal in the frequency domain, such as in Equation (Equation 2).
(2)$$\hat{S}\left[f\right]=X\left[f\right]-\hat{N}\left[f\right],$$
where *f* is the frequency bin. However, doing this directly may insert artifacts (or as Berouti put it: an “annoying musical noise”) if the resulting subtraction has a magnitude is below zero. To counter this, Berouti [16] proposed to carry out this subtraction based on the frequency-wise SNR and prevent the subtraction result to go below a spectral floor based on the estimated noise signal. Equation (Equation 3) presents the method to be used, referred to here as Non-Linear Noise Subtraction (NLNS).
(3)$$\hat{S}\left[f\right]=\left\{\begin{array}{cc} X\left[f\right]-\alpha \left[f\right]\hat{N}\left[f\right], & \mathrm{if}\;X\left[f\right]-\alpha \left[f\right]\hat{N}\left[f\right]>\beta \hat{N}\left[f\right]\;\\ \beta \hat{N}\left[f\right], & \mathrm{otherwise} \end{array}\right.,$$
where 0<β<<1 is a value that establishes the presence of the spectral floor; and α[f] is selected based on the estimated frequency SNR as presented in Equation (Equation 4).
(4)$$\begin{array}{c} \alpha \left[f\right]=\alpha_{max}-SNR_{min}\frac{\alpha_{max}-\alpha_{min}}{SNR_{max}-SNR_{min}}\\ +\frac{X[f]}{\hat{N}\left[f\right]}\frac{\alpha_{max}-\alpha_{min}}{SNR_{max}-SNR_{min}} \end{array},$$
where $\alpha_{max}$ and $\alpha_{min}$ are the maximum and minimum values of α, respectively; and $SNR_{max}$ and $SNR_{min}$ are the maximum and minimum SNR values, respectively.

Typically, the estimated noise $\hat{N}$ is considered as the mean value of the first windows of the recorded signal. However, in this evaluation, the mean-energy-per-frequency of the spectrogram of Recording 6 (as already shown in Figure 5b) is used for this purpose.

Additionally, in our tests the following values were used and chosen based on recommended values obtained from (Equation 2): $\alpha_{min}=1$, $\alpha_{max}=3$, $SNR_{min}=5$, $SNR_{max}=20$, and β=0.3.

### 4.2. Adaptive Quantile Based Noise Estimation (AQBNE)

It can be assumed that the energy of each frequency band is at the noise level a *Q*-th amount of time [21]. This amount of time can be represented as quantile of the ordered distribution of the energy values. By specifying beforehand the value of the *Q* percentile, the energy of the noise signal at each frequency can be calculated as in Equation (Equation 5).
(5)$$\hat{N}\left[f\right]=D_{f}\left[Q\right],$$
where 0<Q<1 is the pre-specified quantile; and $D_{f}$ is the energy distribution in frequency *f* from a pre-specified amount of time $t_{past}$.

This can be pushed even further by estimating a *Q* per frequency ($Q_{f}$) [17]. It does this by calculating the intersection between $D_{f}$ and a *Q*-estimation curve *K* defined in Equation (Equation 6).
(6)$$K\left(Q_{f}\right)=e^{(Q_{min}-Q_{f})\tau },$$
where τ is the slope of the curve; and $Q_{min}$ is the minimum value of *Q*. Both these values need to be pre-specified. Once the noise signal is estimated, NLNS is applied for spectral subtraction; thus, β also requires being pre-specified.

To choose the values of $Q_{min}$, $t_{past}=0.25$, τ and β (for the NLNS part of the filter), a series of experiments were carried out with clean signals of the environment and artificially inserting noise. In this way, an accurate estimation of the SNR could be calculated. A series of values were tested, and the best SNR was obtained with Q=0.29, $t_{past}=0.025$, τ=4 and β=0.01. The rest of the values required for the NLNS part of this filter were set as described in Section 4.1.

### 4.3. Improved Minima Controlled Recursive Averaging (IMCRA)

The first version of this filter estimated the noise by averaging past energy values over each frequency band, and then weighted over the probability of the presence of speech. The speech probability is calculated based on the minima values of energy, similar to AQBNE, however in this case these values are extracted from the smooth spectrogram.

This filter was later improved upon [18]. It divided the process into two stages:
**Smoothing.** Its objective is to provide a rough frequency-wide voice activity detection via the original algorithm.**Minimum Tracking.** Its objective is to carry out a secondary smoothing, but avoiding areas where speech energy is present.

This process results in making the minimum tracking robust and less variant. However, this makes the computation of the speech presence probability biased toward big values. The authors, thus, provide a bias compensation factor in their noise estimator.

Unfortunately, a detailed explanation of this filter requires much more space than can be adequately allocated in this work, and it is outside the scope of this paper. However, it is important to mention that there are 9 parameters that are part of the calibration of this filter. The authors themselves provide a set of recommended values for each parameter. However, as it will be seen in Section 5.2, this set of recommended values did not provide good results. To counter the possibility of inappropriate calibration, we carried out a similar set of experiments as in Section 4.2 to choose the parameters of this filter. Thus, the results for this filter will be presented with two different configurations: the ‘default’ configuration presented in [18] and our ‘alternate’ configuration that was obtained from our preliminary experiments. The values for each configuration are presented in Table 4.

## 5. Evaluation and Results

There were two evaluations that were carried out, each with a separate objective: (1) to measure the amount of ego-noise that was removed or filtered/canceled out, and (2) toevaluate if the noise signature of the other drone flying in parallel (3DR or Parrot) is collaterally extracted.

### 5.1. Ego-Noise Reduction

The three filters described in Section 4 were applied to Recording 9 to measure their effects on the Matrice’s ego-noise while flying statically. Recording 6 also presents the Matrice flying statically, however, because the noise estimation for NLNS is extracted from this recording, using Recording 9 presents a fairer comparison to the other two filters. These results are shown in Table 5.

To measure the amount of ego-noise removal when the drone is moving, Recordings 5 (moving vertically) and 7 (moving horizontally) were used. Table 6 and Table 7 show the reduction of the ego-noise from Recordings 5 and 7, respectively.

Additionally, in Figure 10, Figure 11 and Figure 12, the spectrogram of the original recordings, as well as the filtered from all four configurations, are shown from Recordings 9, 5, and 7. Further discussion of these results is provided in Section 5.3.

### 5.2. Observed Drone Signature Corruption

To evaluate if the signatures of the drones flying in parallel to the Matrice are kept intact in the recordings, the mean-energy-per-frequency of the spectrogram of Recordings 16 and 19 are used for the 3DR and Parrot, respectively. In these recordings, shown in Figure 6a,b, the Matrice is turned off so only the noise signatures of the other drones are recorded. Recordings 18 and 21, were the Matrice and the 3DR or the Parrot are recorded flying in parallel, respectively, were used to evaluate if the filters were able to filter out the ego-noise of the matrice but leave intact the noise signature of the 3DR or the Parrot, respectively. In Figure 13 and Figure 14, the mean-energy-per-frequency of the spectrogram of the resulting filtered spectrogram are shown for all filters, as well as the original noise signature from Recording 16 and 19, respectively.

To evaluate in a more objective manner, the correlation was measured between the noise signatures in Recordings 16 and 19 and the filtered results of Recording 18 and 21, respectively. These correlation results are shown in Table 8 and Table 9.

### 5.3. Results Discussion

As it can be seen in Table 5, the alternate configuration of IMCRA carries out the biggest noise reduction when flying statically, but AQBNE is not that far behind. However, it can be seen from Table 6 and Table 7 that for the rest of the cases, AQBNE removes more ego-noise. Additionally, a visual inspection of the spectrograms of the results shown in Figure 10, Figure 11 and Figure 12, as well as from subjective hearings of the filtered recordings, it is clear that IMCRA in both configurations inserts a far greater amount of artifacts than AQBNE.

Additionally, it can be seen from Figure 13a and Figure 14, that there is still a clear amount of presence of the Matrice noise signature in the filtered results (which can be seen in Figure 5b). If the high correlation factors seen in Table 8 and Table 9 are also considered, both of these results imply that the tendencies of the noise signatures of the three drones are very close to each other. This is further evidence at what was suggested from the correlation between the noise signatures of the three drones presented in Table 2.

All of this in conjunction implies that, even though AQBNE is the preferable filter from the three chosen in terms of ego-noise removal, it is not able to keep intact the noise signature of the other drone flying in parallel. This may not be a fault of the filter itself, however, since both signatures are very correlated, and their removal may require a more sophisticated method of source separation.

It is also important to mention that the configuration process of these filters may be improved, which may reduce the presence of artifacts.

## 6. Conclusions and Future Work

In this paper, an in-depth analysis of the AIRA-UAS corpus was carried out. It was found that: The environment in which it was recorded had a very small amount of noise; the current drones’ noise signatures may not be sufficient for discrimination, and they have high correlation between them, making their extraction challenging; the ego-noise is dynamic, having a bigger variance when the drone is moving horizontally than when moving vertically or flying statically; and the recordings have a severely low SNR which, again, will make the noise signature extraction very challenging. This was further evidenced when the three popular filtering techniques that were evaluated did not provide convincing results.

The filtering techniques that were evaluated were Berouti’s Non-Linear Noise Subtraction, Adaptive Quantile Based Noise Estimation, and Improved Minima Controlled Recursive Averaging. The latter required two configurations of its parameters to be tested since its default configuration delivered poor performance against the low SNR.

In terms of noise reduction, AQBNE provided the best results, with the proposed alternate configuration of IMCRA coming in close second when the drone is not moving. IMCRA provided a severe amount of artifacts in both configurations. To be fair, however, other configurations of this filter may counter this, which is left for future work.

These results make it clear that there is a prime field of research in audio processing over drones since current popular algorithms are not up to par for its challenge, opening the door to investigate alternative methods to tackle this task.

To this effect, other type of audio processing algorithms will be evaluated in future works, such as beam-forming techniques and source separation techniques. Geometric Source Separation, for example, is a good candidate for further evaluation.

It is important to consider that the ego-noise of the drone is dynamic but not unpredictable. Since the control signal of the drone’s motor is known, a model could be created the uses this information to predict the noise emitted by the drone and provided a better noise estimation to be canceled. The implementation of this algorithm is also left for future work.

Finally, to overcome the issue of high correlation between the drones’ noise signatures, a drone can be designed with this in mind such that its noise signature does not have such a high correlation with other drones. However, this would require a new rotor design, as well as a new propeller design, and as such is left for future work, as well.

## Figures and Tables

**Figure 1 sensors-19-03902-f001:**
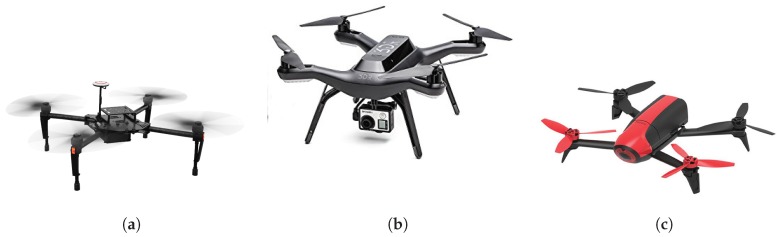
Drones used. (**a**) DJI Matrice 100, ©2018 IEEE [12]; (**b**) 3DR Solo, ©2018 IEEE [12]; (**c**) Parrot Bebop 2, ©2018 IEEE [12].

**Figure 2 sensors-19-03902-f002:**
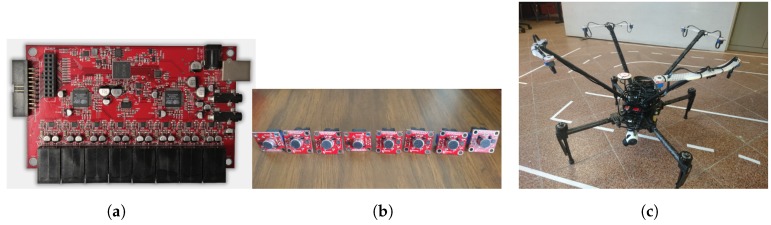
Audio capture system. (**a**) 8SoundsUSB audio interface, ©2018 IEEE [12]; (**b**) The microphones used, ©2018 IEEE [12]; (**c**) Complete capture system, ©2018 IEEE [12].

**Figure 3 sensors-19-03902-f003:**
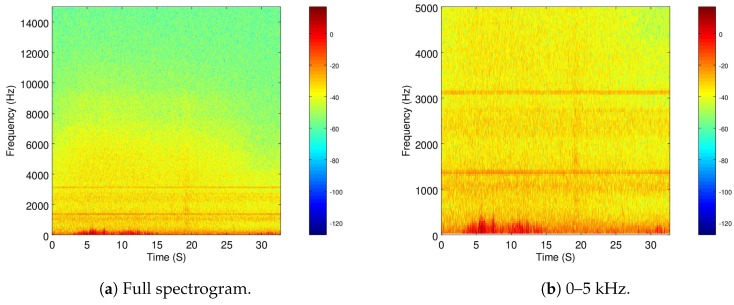
Time-frequency spectrogram of Recording 2 (background noise).

**Figure 4 sensors-19-03902-f004:**
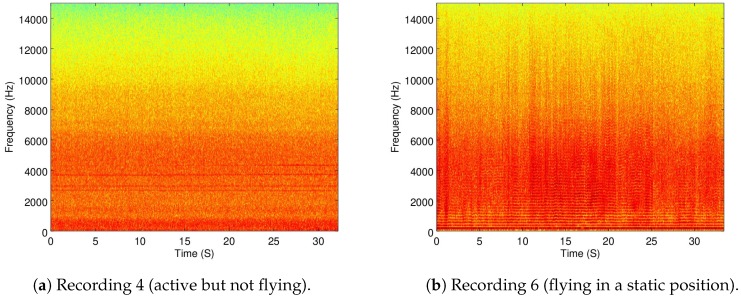
Time-frequency spectrograms of Recordings 4 and 6.

**Figure 5 sensors-19-03902-f005:**
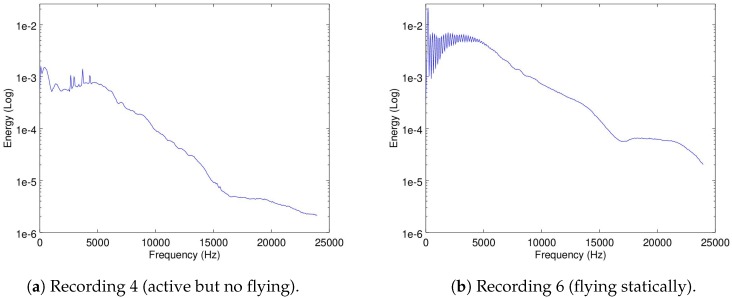
Mean-energy-per-frequency of the spectrograms of Recordings 4 and 6.

**Figure 6 sensors-19-03902-f006:**
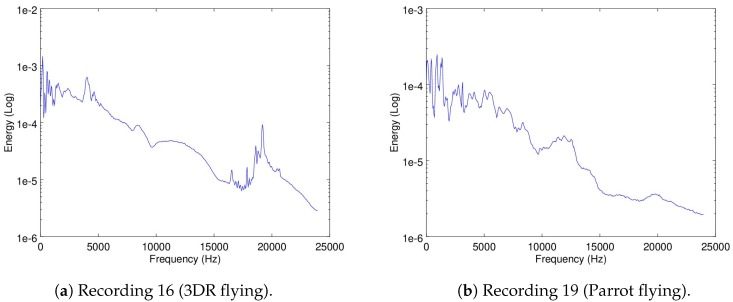
Mean-energy-per-frequency of the spectrograms of Recordings 16 and 19.

**Figure 7 sensors-19-03902-f007:**
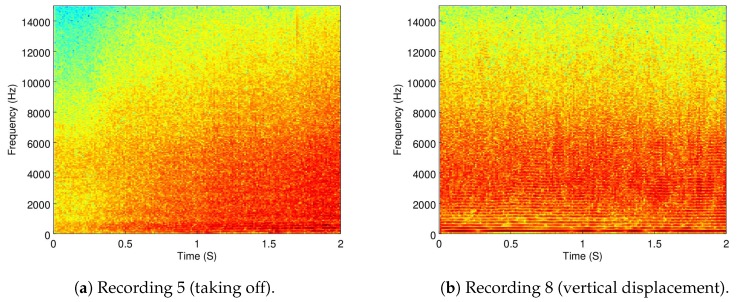
The 0–2 s zoomed spectrograms of Recordings 5 and 8.

**Figure 8 sensors-19-03902-f008:**
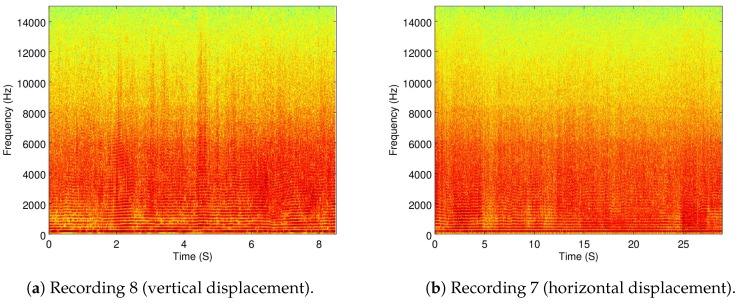
The spectrograms of Recordings 8 and 7.

**Figure 9 sensors-19-03902-f009:**
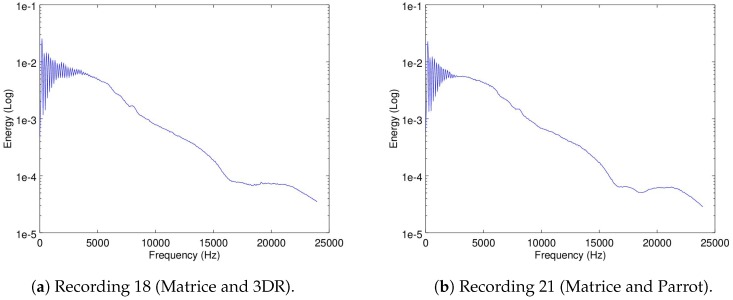
Mean-energy-per-frequency of spectrograms of Recordings 18 and 21.

**Figure 10 sensors-19-03902-f010:**
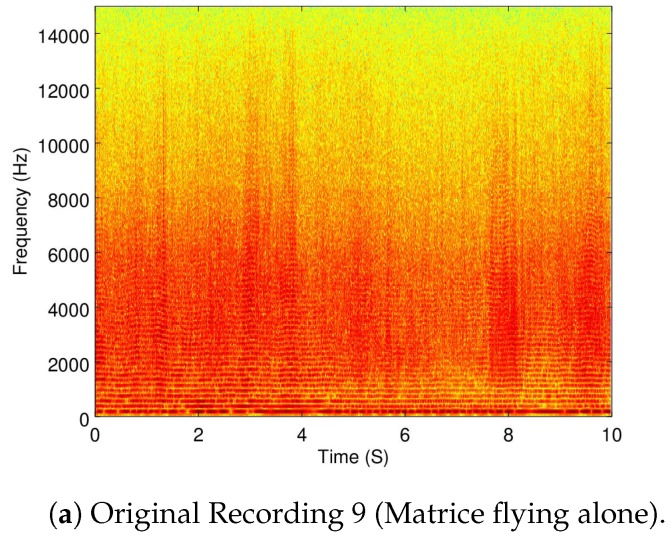
Filtering results of Recording 9 (Matrice flying alone).

**Figure 11 sensors-19-03902-f011:**
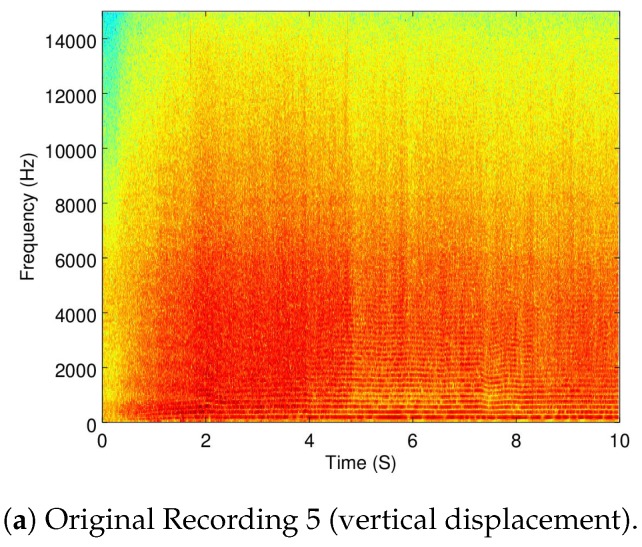
Filtering results of Recording 5 (vertical displacement).

**Figure 12 sensors-19-03902-f012:**
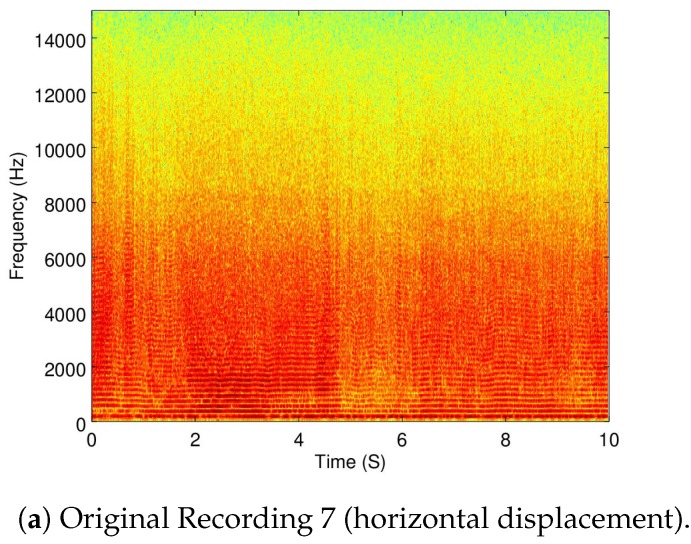
Filtering results of Recording 7 (horizontal displacement).

**Figure 13 sensors-19-03902-f013:**
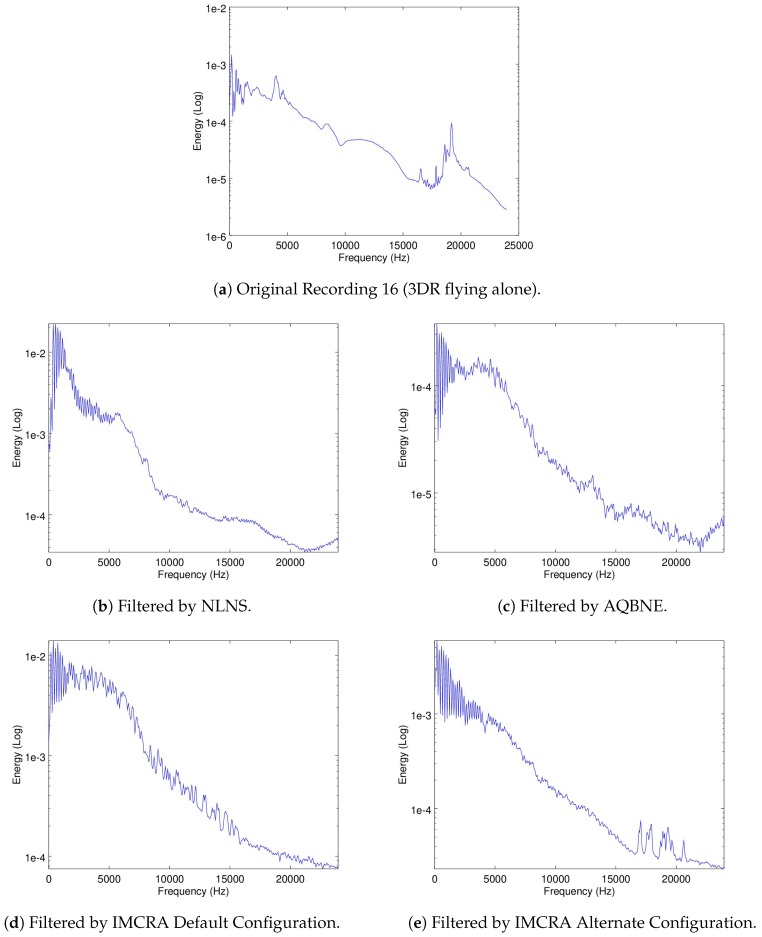
Filtering results of Recording 18 (both Matrice and 3DR flying side-by-side).

**Figure 14 sensors-19-03902-f014:**
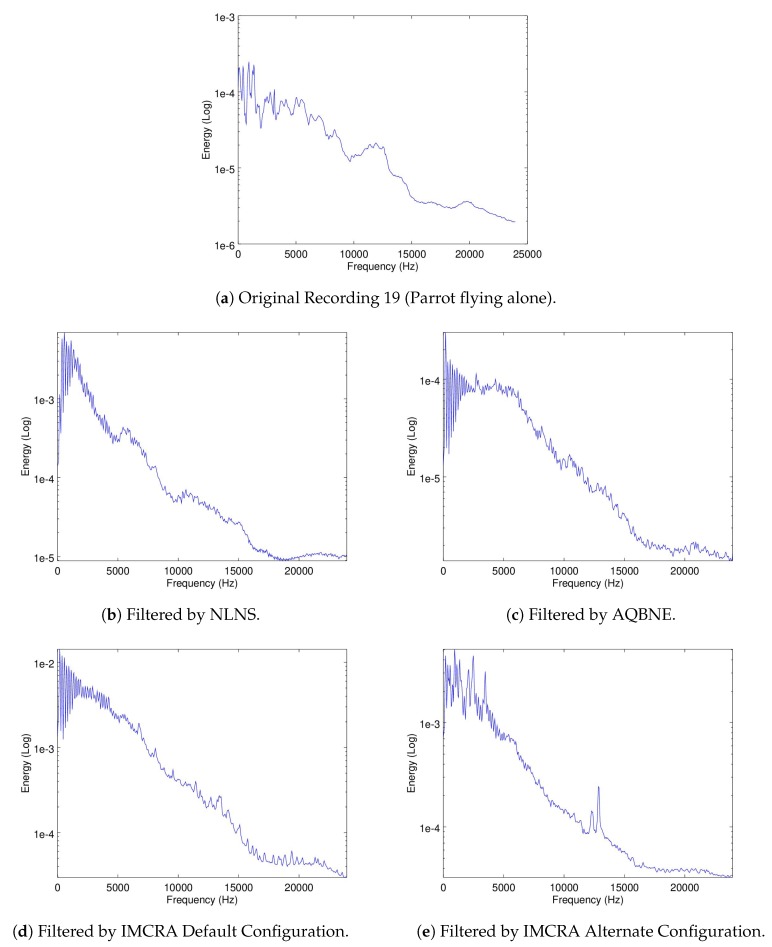
Filtering results of Recording 21 (both Matrice and Parrot flying side-by-side).

**Table 1 sensors-19-03902-t001:** Details of the recording settings of 1st protocol with the DJI Matrice 100.

Recording #	Drone Activated?	Horizontal Vel [m/s]	Vertical Vel. [m/s]	Const. Height [m]
1	NO	0	0	0
2	NO	0	0	0
3	NO	0	0	0
4	YES	0	0	0
5	YES	0	0.6	0–3
6	YES	0	0	3
7	YES	1.4	0	3
8	YES	0	0.4	3–5
9	YES	0	0	5
10	YES	1.4	0	5
11	YES	0	0.6	5–10
12	YES	0	0	10
13	YES	1.4	0	10
14	YES	0	0	15
15	YES	0	0	20

**Table 2 sensors-19-03902-t002:** Correlation between all noise signatures.

	Matrice	3DR	Parrot
**Matrice**	1.00	0.94	0.81
**3DR**	0.94	1.00	0.83
**Parrot**	0.81	0.83	1.00

**Table 3 sensors-19-03902-t003:** Frequency-wise average of the energy variance per frequency of Recordings 6, 8, and 7.

	Recording 6 Static Position	Recording 8 Vertical Displacement	Recording 7 Horizontal Displacement
Energy Variance	$2.7077\times 10^{-6}$	$2.9369\times 10^{-6}$	$3.3799\times 10^{-6}$

**Table 4 sensors-19-03902-t004:** The default and alternate configuration to be evaluated of Improved Minima Controlled Recursive Averaging (IMCRA).

	$\alpha_{s}$	*U*	*V*	$B_{min}$	$\gamma_{0}$	$\gamma_{1}$	$\zeta_{0}$	$\alpha_{d}$	β
**Default Configuration**	0.9	8	15	1.66	4.6	3	1.67	0.85	1.47
**Alternate Configuration**	0.9	6	11	3.16	7.6	3	10.67	0.85	50

**Table 5 sensors-19-03902-t005:** Ego-noise reduction in dBFS for Recording 9 (flying statically). NLNS = Non-Linear Noise Subtraction; AQBNE = Adaptive Quantile Based Noise Estimation.

	NLNS	AQBNE	IMCRA Def.	IMCRA Alt.
Before	−25.60	−25.60	−25.60	−25.60
After	−42.53	−56.13	−33.23	−64.42
Difference	16.93	30.53	7.63	38.82

**Table 6 sensors-19-03902-t006:** Ego-noise reduction in dBFS for Recording 5 (moving vertically).

	NLNS	AQBNE	IMCRA Def.	IMCRA Alt.
Before	−21.92	−21.92	−21.92	−21.92
After	−27.72	−50.38	−22.73	−28.92
Difference	5.80	28.46	0.80	7.00

**Table 7 sensors-19-03902-t007:** Ego-noise reduction in dBFS for Recording 7 (moving horizontally).

	NLNS	AQBNE	IMCRA Def.	IMCRA Alt.
Before	−23.57	−23.57	−23.57	−23.57
After	−32.57	−54.24	−28.53	−41.57
Difference	9.00	30.67	4.96	18.00

**Table 8 sensors-19-03902-t008:** Correlation between Recording 16 (3DR flying alone) and filtered results from Recording 18 (Matrice and 3DR flying in parallel).

	NLNS	AQBNE	IMCRA Def.	IMCRA Alt.
Correlation	0.69	0.94	0.91	0.88

**Table 9 sensors-19-03902-t009:** Correlation between Recording 18 (Parrot flying alone) and filtered results from Recording 21 (Matrice and Parrot flying in parallel).

	NLNS	AQBNE	IMCRA Def.	IMCRA Alt.
Correlation	0.69	0.84	0.84	0.85

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
