# Peer review of "On the Use of the AIRA-UAS Corpus to Evaluate Audio Processing Algorithms in Unmanned Aerial Systems†"

_sensors, 2019, doi:10.3390/s19183902_

Round 1
Reviewer 1 Report
This work addresses an interesting and relevant problem. There is currently a huge interest in the problem, from many different perspectives. Even though the introduction is reasonably well written, the broader view was not represented very well. For example, there exists a huge variety of noise cancellation methods and techniques in the literature, but the authors chose to evaluate only three of them. The reason for this choice was not strong enough to make a case. In my opinion, the authors must provide a stronger justification for that choice.
Specific Points to improve:
Use of acronym UAV in the Abstract is unnecessary. Keywords list must be improved: for example "unmanned" should be changed to "Unmanned Aerial Systems", or "Unmanned Aerial Vehicles". Parenthesis missing when referencing equations in the main text. Explicit website link in the text should be replaced with references (see how to reference websites using BibTeX). Figures must be improved. Most are too small, and lack x and y labels. This is not acceptable in a technical work at this level. Figure captions must be improved as well. Most cases, caption texts are not explicative enough.
Overall presentation of results and analysis: could have been done better. Figures presented should reflect main results, and description should be provided for other cases that the authors think should be highlighted. I also recommend the authors to improve the overall composition, especially with regards to the presentation of results and discussion.
Reviewer 2 Report
The authors show a novel application of UAV focused on audio processing over drones. The research is very interesting and well developed, but some aspects should be improved before publication, specially those related to the results depicting.
Figures should indicate magnitude and units in axis. In addition, axis should be big enough to be easily readable and color information in figures should be explained with a colormap legend. Some examples are Figure 7- 10. Figures 11, 12, etc does not present any information on axis... Please improve it.
Reviewer 3 Report
# Remarks regarding content:
In general:
Newer references from 2018 and 2019 are missing, except for own ones (ref. 8+9).
194:
Vertical displacement in the video 6 sec (3:51 – 3:57) but the recording 8 is over 8 sec and also the spectrogram (fig. 17) shows a time interval of about 8 sec. => Pls. check the data set, maybe it contains a hovering part.
Furthermore, according to the video, the data set of the horizontal displacement contains a part, in which the UAV hovers and returns at the end of the recording. (Thus, it’s no data of a uniform horizontal displacement.)
197 - 199:
Not enough data for this conclusion; a statistical analysis is needed. Further factors (e.g. wind, sensor and controlling accuracy) were not considered. See also the note referring to line 192.
Fig. 15/16:
Caption says “… 0-5 kHz …” but spectrogram shows the frequency range 0-24 kHz.
# Visual presentation:
Spectrograms
Please, scale up all diagrams, especially the axis labels and scaling numbers. Please, add a color scale and physical unity to the axis labels, e. g. “Frequency [Hz]”.
Mean-energy-per-frequency plots
Please scale up the diagrams / scaling numbers and add axis labels. A logarithmic scaling and a limitation to 15 kHz would be better to see/compare details.
# Minor issues and typos
71 „to provided“ => to provide
144 “These analysis are” => These analyses are
172 “laying in the ground.” => laying on the ground.
189 delete “each recordings”
215 “poinpoint” => pinpoint
222 “pickup” => pick up
223 “possiblity” => possibility
227 “depending the on the motor activity” => depending on the motor activity
275 “spectogram” => spectrogram
279 “to carried out” => to carry out
327 “from Figures Figures 36 to 45” => from Figures 36 to 45
Fig. 7/8 The spectrogram looks like recording 2 (channel 1). => Therefore, the caption of figure 7 should be "... spectrogram of Recording 2".
Fig. 8 The label of the time axis is missing.
